# TOWARDS POWERFUL GRAPH NEURAL NETWORKS: DIVERSITY MATTERS

## ABSTRACT

Graph neural networks (GNNs) offer us an effective framework for graph representation learning via layer-wise neighborhood aggregation. Their success is attributed to their expressive power at learning representation of nodes and graphs. To achieve GNNs with high expressive power, existing methods mainly resort to complex neighborhood aggregation functions, e.g., designing injective aggregation function or using multiple aggregation functions. Consequently, their expressive power is limited by the capability of aggregation function, which is tricky to determine in practice. To combat this problem, we propose a novel framework, namely *diverse sampling*, to improve the expressive power of GNNs. For a target node, diverse sampling offers it diverse neighborhoods, i.e., rooted sub-graphs, and the representation of target node is finally obtained via aggregating the representation of diverse neighborhoods obtained using *any* GNN model. High expressive power is guaranteed by the diversity of different neighborhoods. We use classical GNNs (i.e., GCN and GAT) as base models to evaluate the effectiveness of the proposed framework. Experiments are conducted at multi-class node classification task on three benchmark datasets and multi-label node classification task on a dataset collected in this paper. Extensive experiments demonstrate the proposed method consistently improve the performance of base GNN models. The proposed framework is applicable to any GNN models and thus is general for improving the expressive power of GNNs.

## 1 INTRODUCTION

Graph neural networks (GNNs) have been shown to be effective at graph representation learning and many predictive tasks on graph-structured data, e.g., node classification and graph classification (Kipf & Welling, 2016; Xu et al., 2018a). GNNs follow a neighborhood aggregation scheme, where the representation of a node is obtained by recursively aggregating and transforming representation of its neighboring nodes (Gilmer et al., 2017). The success of GNNs is believed to be attributed to their high expressive power at learning representation of nodes and graphs (Xu et al., 2018a). Therefore, it is an important research problem to analyze and improve the expressive power of existing GNN models and design new GNNs with high expressive power.

Several recent works focus on the expressive power of GNNs. Xu et al. pointed out that the expressive power of GNNs depends on the neighborhood aggregation function (Xu et al., 2018a). They develop a simple architecture, i.e., leveraging multi-layer perceptron (MLP) and a sum pooling as a universal approximator defined on multi-set, to achieve injective neighborhood aggregation function. With injective aggregation function in each layer, the proposed graph isomorphism network (GIN) has the expressive power as high as the Weisfeiler-Lehman (WL) graph isomorphism test (Weisfeiler & Lehman, 1968). Similarly, Sato et al. implement a powerful GNN via consistent port numbering, i.e., mapping edges to port numbering and neighbors are ordered by the port numbering (Sato et al., 2019). However, port ordering of CPNGNNs is not unique, and not all orderings can distinguish the same set of graphs (Garg et al., 2020). Principal neighborhood aggregation (PNA) defines multiple aggregation functions to improve the expressive power of GNNs (Corso et al., 2020). However, the number of required aggregation functions to discriminate multi-sets depends on the size of multi-set, which is prohibitive for real world networks with skewed degree distribution. In sum, existing methods focus on designing an injective, often complex, aggregation function in each layer to achieve GNNs with high expressive power. However, injective functions are difficult to obtain and tricky to

determine in practice. Indeed, layer-wise injective function is not always required and what we need is an injective function defined over rooted sub-graphs or graphs as a whole.

In this paper, we propose a novel framework, namely diverse sampling, to improve the expressive power of GNNs. For a target node, diverse sampling offers it diverse neighborhoods, i.e., rooted sub-graphs, and the representation of target node is finally obtained via aggregating the representation of diverse neighborhoods obtained using any GNN model. High expressive power is guaranteed by the diversity of different neighborhoods. For convenience, we denote with DS-GNN the GNN implemented under the proposed diverse sampling framework.

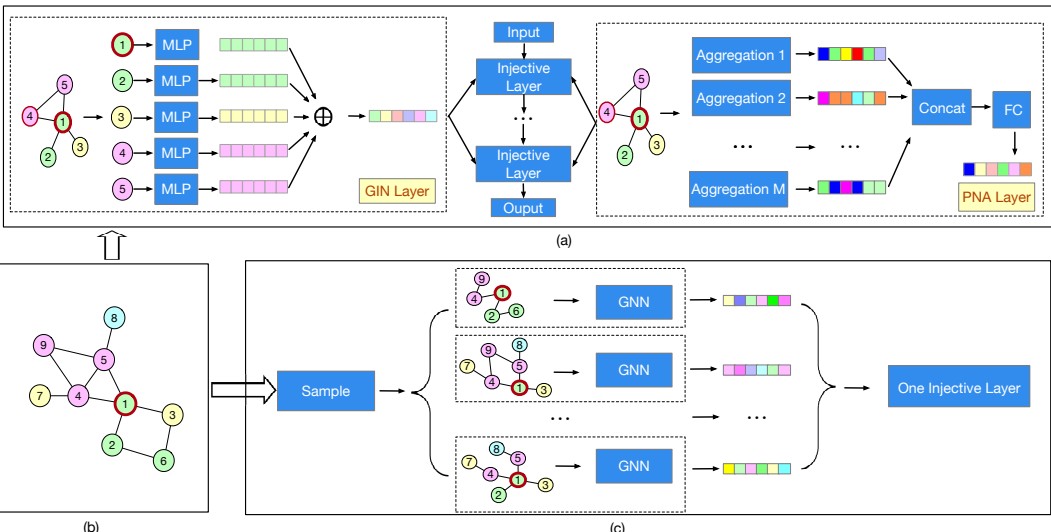

Figure 1: The motivation of DS-GNN: constructing multiple sampled graphs rather than complex layer-wise aggregation functions (the node with red circle is the central node and "FC" represents fully connected layer).

Fig. 1 illustrates the main idea of the proposed DS-GNN, and compare it with two representative methods, i.e., GIN and PNA. Fig. 1 (a) depicts the injective layer implemented via MLP or multiple aggregation functions, aggregating first-order neighboring nodes to obtain the representation of central node. Injective layer are stacked to achieve an overall injective function defined on rooted sub-graphs. On the contrary, DS-GNN does not follow the line of designing complicated aggregation functions in each layer. Instead, DS-GNN improve the expressive power of GNNs via obtaining diverse rooted sub-graphs for each node. Specifically, we sample nodes multiple times on the entire input graph based on diverse sampling, and obtain multiple sampled sub-graphs for each node. After diverse sampling, we leverage the shared GNN model to get the representation for the central node, including its high-order neighbors. In this way, each node is represented by a multi-set, consisting of the representations obtained from different sampled rooted sub-graphs. The final representation of central node is finally obtained via aggregating the representation of diverse neighborhoods.

Finally, we use classical GNNs (i.e., GCN (Kipf & Welling, 2016) and GAT (Veličković et al., 2017)) as base models to evaluate the effectiveness of the proposed framework. Experiments are conducted at node-based multi-class classification task on three benchmark datasets and node-based multi-label classification task on a dataset collected in this paper. Extensive experiments demonstrate the proposed method consistently improve the performance of base GNN models. The proposed framework is applicable to any GNN models and thus is general for improving the expressive power of GNNs.

## 2   NOTATIONS AND PRELIMINARIES

We first introduce the general framework of GNNs.

$G = \{V, A\}$ denotes an undirected graph, where $V$ is the set of nodes with $|V| = n$, and $A$ is the adjacency matrix with $A_{i,j} = A_{j,i}$ to define the connection between node $i$ and node $j$. $X \in R^{n \times p}$ denotes the feature matrix and the $i$-th row in $X$ represent the attributes of $i$-th node. DS-GNN Modern GNNs follow a neighborhood aggregation scheme, which iteratively update each node's representation via aggregating the representation of its neighboring nodes. Formally, the $k$-th layer of GNN is

$$a_v^{(k)} = \text{AGGREGATE}^{(k)}(h_u^{(k-1)} : u \in N(v)), \qquad h_v^{(k)} = \text{COMBINE}^{(k)}(h_v^{(k-1)}, a_v^{(k)}), \quad (1)$$

where $N(v)$ represents the neighbors of node $v$, $h_v^{(k)}$ is the representation of node $v$ in the $k$-th layer, and $h_v^{(0)} = X_v$. Additionally, we introduce two representative base models: Graph Convolutional Network (GCN) (Kipf & Welling, 2016) and Graph Attention Networks (GAT) (Veličković et al., 2017) in Appendix A.

## 3    DIVERSE SAMPLING BASED POWERFUL GRAPH NEURAL NETWORK

Considering that the injectivity of each node with its neighbors making GNNs be possessed with the most powerful ability for node-level task, i.e., representations can be distinguishable for two nodes when they have dissimilar attributes or different neighbors, we devote to designing a powerful GNN to reach such injectivity, thus increase its expressive power. Specifically, we propose a novel framework, i.e., Diverse Sampling (DS-GNN), to increase the expressive power via constructing diverse rooted sub-graphs for each node. In this section, we first give the framework of our DS-GNN. Then we will introduce the important parts of DS-GNN, including diverse sampling strategy that ensure the diversity among each sampling, as well as the part of model learning. We also theoretically analyze how the proposed DS-GNN can improve the expressive abilities when added to generic GNNs.

### 3.1    METHOD

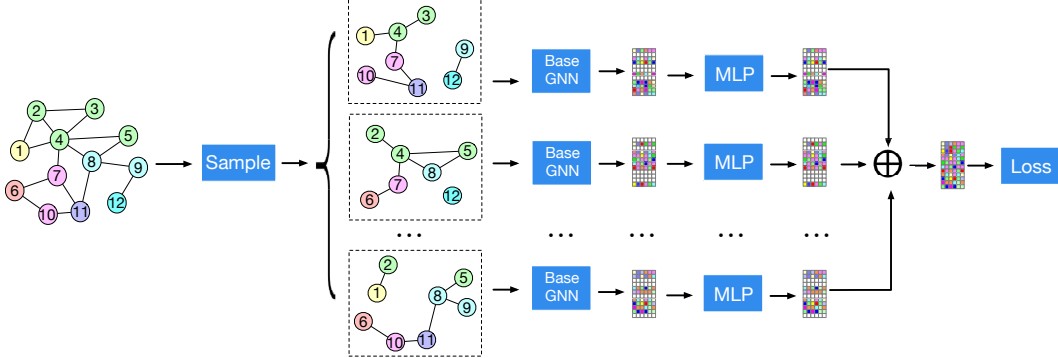

Figure 2: Architecture of DS-GNN.

The framework of DS-GNN is illustrated in Fig. 2. For the input graph, instead of running GNN on the entire graph, we first do node sampling with $K$-times and then obtain $K$ sampled graphs. During sampling, we calculate the sampling probability for each node and randomly retain the node based on the probability. If one node is not sampled, it means that the node and all edges connected with it will be removed from the graph. The updated adjacency matrix and feature matrix of each sampling are fed into the base GNN model, and the GNN is shared over all samplings. Then we obtain one representation for each sampled node corresponded with one input sampled graph. Thus, if one node is sampled more than one time, we will obtain multiple representations.

To get the final representation of each node, we need to integrate these representations. To hold the injectivity when integrating representations, we adopt a multi-layer perceptrons (MLP) followed by the sum aggregation to achieve the injective multi-set aggregation function. Once getting the final representation for each node, we calculate the loss function and optimize the model.

**Sampling Strategy**

Intuitively, with a large number of sampling numbers, the sampling results are expected to retain the information of entire graph. Thus previous methods, e.g., Dropedge (Rong et al., 2019) which samples in each training epoch, do not need to carefully design the sampling strategy and they keep all edges when testing and validation. Meanwhile, they fail to achieve diverse sampled graphs in testing. Different from previous methods, we need to hold the $K$ sampled graphs in testing and keep training and testing consistently. To solve this, we only sample the entire graph $K$-times, and leverage these sampled $K$ graphs in all training epochs, as well as the validation and testing. To guarantee the performance of the proposed DS-GNN when even small K is chosen, we carefully design the important diverse sampling strategy.

To ensure the reuse of sampling results, we propose node sampling to obtain $K$ sampled graphs. In one sampling, the neighbors of each node are obtained from the same sampled graph. For the sampling probability of nodes, the most intuitive way is to set the same initial probability for each node, which is $p_{init}$. However, this method may require a large number of sampling numbers $K$. In order to further speed up or get an effective sampling plan, we also propose three intuitive guidelines for sampling strategy: 1) to ensure the coverage of samples, each node should appears at least in one sampled graph; 2) to achieve the diversity among samples, we assume that the sampling probability of each node should decrease when its sampled times in previous is already large; 3) to retain the important node in the original network and adopting the node degree as the importance indicator, we assume that the sampling probability of each node should increase when its degree in original network is large. Meanwhile, we also considering the previous sampled results, discounting the sampling probability of each node when the its degree in previous sampled graphs is already large. That's to say, the sampling probability of each node increase with the increase of node degree in original network, and decrease with the increase of node degree in previous sampled graphs. To achieve the above three guidelines, we need to relate the sampling probability of each node to the historical sampling results. Next, we will introduce in detail how we do.

Let $K$ represent the total number of samples, $p_{v,i}$ represent the probability of node $v$ in the $i$-th sampling, and $H_{v,i}$ represent the number of times that the node $v$ has been sampled before the $i$-th sampling. To achieve guideline 1), we should guarantee that the probability for a node, which has never been sampled in history $K$-1 samples, equals to 1 at the $K$-th sampling. In other words, $p_{v,K} = 1$ when $H_{v,K} = 0$. Additionally, to achieve the diversity guideline of 2), the sampling probability $p_{v,i}$ should decrease with the increase of historical sampled numbers $H_{v,i}$. Together, the sampling probability of node $v$ for the $i$-th sampling is defined as:

$$p_{v,i} = p_{init} + (1 - p_{init}) \times \frac{i - H_{v,i}}{K}. \tag{2}$$

To achieve guideline of 3), a new sampling strategy is designed. Specifically, we define $D_{v,j}^{sample}$ as the degree of node $v$ in the j-th sampled graph, and $D_v$ as the degree of node $v$ in the original graph. To make the sampling probability increase with $D_v$, as well as decrease with $D_{v,j}^{sample}$ as mentioned in guideline 3), we define the coefficient $D_{v,i}^{gap}$ as:

$$D_{v,i}^{gap} = \frac{\sum_{j=1}^{i-1}(D_v - D_{v,j}^{sample})}{\sum_{j=1}^{i-1} D_v} \tag{3}$$

To achieve our three guidelines in total, we integrate the Eq 2 and Eq 3, and define the sampling probability for node $v$ in the $i$-th sampling as:

$$p_{v,i} = p_{init} + (1 - p_{init}) \times \frac{i - H_{v,i}}{K} \times D_{v,i}^{gap}. \tag{4}$$

Note that our sampling strategy is different from previous methods, i.e., dropping features and dropping edges. The Dropout (Srivastava et al., 2014) is proposed to prevent over-fitting by randomly dropping features. Dropping edges can be regarded as a generation of Dropout from dropping features to dropping edges. However, dropping edges may cause that two nodes who are the same with identical neighbors obtain different representations, since dropping edges may cause these two nodes result in different neighbors after edge sampling. In contrast, we propose node sampling strategy, ensuring that the nodes with the same input features and neighbors will still have the same

representation. We give the detailed introduction of some previous methods, included in dropping edges in Appendix B.

**Model Learning**

In the following, we describe in detail how DS-GNN can be applied to any base GNN to achieve a more powerful GNN model. Specifically, we first do $K$ samplings on the input graph $A$ via the above introduced diverse sampling, getting the sampled graphs $[A_1, A_2, \cdots, A_K]$ in a pre-processing step. Then we applied a base GNN model to obtain the representations of nodes in sampled graphs respectively, each layer is formulated as.

$$h_v^{(k,l)} = \text{COMBINE}^{(l)}(h_v^{(k,l-1)}, \text{AGGREGATE}^{(l)}(h_u^{(k,l-1)} : u \in N_k(v))), \tag{5}$$

where $h_v^{k,l}$ is the representation of node $v$ at the $l$-th layer in the $k$-th sampled graphs, and $N_k(v)$ denotes the neighbors of node $v$ in the $k$-th sampled graphs $A_k$. For each node, we obtain $K$ representations $\{h_v^{(1,L)}, h_v^{(2,L)}, \cdots, h_v^{(K,L)}\}$ from the $L$-th layer of base GNN based on the $K$ sampled graphs. Note that if a node is not sampled in one sampling, we fill its representation with 0 to ensure that each node has $K$ representations. In order to integrate them into one representation, we fed them into a MLP with two fully connected layers, and then sum the transferred representations after MLP. Note that the MLP is shared over all samplings. Let $O_v$ denote the output representation of node $v$, $O_v$ is calculated as

$$O_v = \sum_{i=1}^{K} \text{MLP}(h_v^{(i,L)}). \tag{6}$$

To verify the ability of DS-GNN model, we adopt node-level tasks, i.e., node based multi-class classification and node-based multi-label classification, as the target task. Note that in multi-label classification, each node has multiple labels. Let $V^{Label}$ denote the set of labeled nodes and $m$ is the number of total labels, $Y_{vc}$ equals 1 if node $v$ has label $c$, otherwise 0. For node-based multi-class classification, the output probability vector $\hat{Z}_v$ is calculated by applying "Softmax" funtion to the output representations $O_v$:

$$\hat{Z}_v = \text{Softmax}(O_v). \tag{7}$$

For node-based multi-label classification, $\hat{Z}_v$ is calculated by applied "Sigmoid" funtion instead:

$$\hat{Z}_v = \text{Sigmoid}(O_v). \tag{8}$$

For these two tasks, the loss function is defined as the cross-entropy over all labeled nodes as

$$\mathcal{L} = - \sum_{v \in V^{Label}} \sum_{c=1}^{m} Y_{vc} \ln \hat{Z}_{vc}, \tag{9}$$

### 3.2 Towards high expressive power

Recall that, in this work we devote to achieving injectivity on nodes to increase the expressive power of GNN, i.e., for any two nodes with different attributes or different neighbors, the model can output different representations for them. In this section, we theoretically analyze whether our proposed DS-GNN method can achieve this goal. Note that the MLP applied to the K representations in Fig. 2 followed by the sum operator can achieve injectivity when integrating the multi-sets (Hornik et al., 1989; Xu et al., 2018a). As a result, to prove the injectivity on nodes of the proposed DS-GNN, we only need to prove that two nodes with different neighbors have different multi-sets of the K representations.

For simplicity, we denote the representation of node $u$ on the $i$-th sampled graph under $L$ layers base GNN model, i.e., $h_u^{(i,L)}$, as $u_i$. Formally, for two nodes, i.e., node $u$ and node $v$, with different attributes or different neighbors on the original graph, the multi-sets of their representations based on the $K$ sampled graphs under $L$ layers base GNN model are:

$$\text{Multi} - \text{set}(u) = \{u_1, u_2, \cdots, u_K\}, \qquad \text{Multi} - \text{set}(v) = \{v_1, v_2, \cdots, v_K\}. \tag{10}$$

We provide the following theorem and prove it in Appendix C to demonstrate that we can obtain different multi-sets for these two nodes with high probability.

**Theorem** As the number of samples increases, the probability that the multi-sets for two nodes with different attributes or different neighbors are different can be close to 1.

On the basis of the high probability of obtaining different multi-sets and the proven injective multi-set function, i.e., MLP followed by "Sum", the injectivity can be achieved among nodes.

Note that for the two nodes who are the same, i.e., they have the same attributes and neighbors, our DS-GNN also ensure that they have the same representation. This is because that we adopt node sampling instead of edge sampling. Under node sampling, the two identical nodes will lose this neighbor at the same time, otherwise, keeping the neighbor, as well as the representations obtained by base GNN model stay the same.

## 4 EXPERIMENTS

We use GCN and GAT as our base models and implement them under our proposed DS-GNN framework respectively, referred as DS-GCN and DS-GAT. The effectiveness of DS-GCN and DS-GAT is evaluated on three benchmark datasets via multi-class classification. Detailed analysis of sampling times and sampling strategy are also provided. In addition, to enrich the type of node-level tasks, we also offer an multi-label classification task on a newly collected dataset from DouBan website[1], namely DBMovie.

### 4.1 EXPERIMENTAL SETTINGS

We use Tensorflow to implement the proposed model and take Adam with an initial learning rate of 0.01 as optimizer. For the three benchmark datasets, we set a weight decay of 0.0005. For DBMovie, we do not use weight decay. Note that for the $i$-th sampling, we sample 10 times and take the average of these 10 sampled garphs as the result of the $i$-th sampling, preventing the instability of a single sampling. All GNN models has two layers and leverage Relu as the activation function of hidden layers. We add a ResNet (He et al., 2016) between the second GNN layer and the output layer. We run 1000 epochs and choose the model that performs the best on validation.

### 4.2 BASELINES

For the three benchmark datasets, GNNs have achieved large improvements than traditional methods. As a result, we choose the the representative GNN models, i.e., GCN (Kipf & Welling, 2016), GAT (Veličković et al., 2017), as the base GNN models and apply our DS-GNN to these two models. Additionally, we compare against the SOTA GNNs, including GraphSAGE (Hamilton et al., 2017), GIN (Xu et al., 2018a) and DropEdge (Rong et al., 2019). For the newly collected DBMovie dataset, we first consider traditional node classification methods as our baselines, including Multi-Layer Perceptrons (MLP), which only use node features, and graph embeddings (DeepWalk) (Perozzi et al., 2014), which only use graph structures. Since GNNs are proved to be effective in graph-based learning, we also compare against GCN and GAT.

Table 1: The Statistics of Datasets

| Datasets | Nodes | Edges | Classes | Features | Train/Validation/Test | Node-level Task |
|----------|-------|-------|---------|----------|----------------------|-----------------|
| Cora | 2,708 | 5,429 | 7 | 1,433 | 1,208/500/1,000 | Multi-class |
| CiteSeer | 3,327 | 4,732 | 6 | 3,703 | 1,812/500/1,000 | Multi-class |
| PubMed | 19,717 | 44,338 | 3 | 500 | 18,217/500/1,000 | Multi-class |
| DBMovie | 21,659 | 221,138 | 28 | 3,000 | 20,159/500/1,000 | Multi-label |

### 4.3 NODE-BASED MULTI-CLASS CLASSIFICATION

To evaluate the proposed method on node-based multi-class classification, we conduct experiments on the three benchmark datasets, namely, Cora, CiteSeer, PubMed. The first three rows in Table 1 show an overview of these three multi-class datasets. To better verify the ability of each model and

---

[1]https://movie.douban.com/

eliminate the effect of other factors such as data insufficiency, we follow the data split in DropEdge (Rong et al., 2019).

**Performance on Node-based Multi-class Classification**

| Method | Cora | CiteSeer | PubMed |
|---|---|---|---|
| GIN | 85.7% | 76.4% | 89.7% |
| GraphSAGE | 87.8% | 78.4% | 90.1% |
| DropEdge-GCN | 86.5% | 78.7% | 91.2% |
| GCN | 86.1% | 75.9% | 90.2% |
| **DS-GCN** | **88.0%** | **79.9%** | **90.5%** |
| GAT | 87.4% | 77.8% | 87.9% |
| **DS-GAT** | **88.2%** | **80.0%** | **91.0%** |

Table 2: Results of Node-based Multi-class Classification

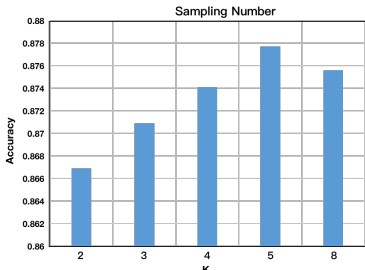

Figure 3: The impact of sampling time K

Experimental results are reported in Table 2. Additionally, we provide the corresponding standard deviation in Appendix D. Similar to previous research, we report the mean classification accuracy over nodes in test set for quantitative evaluation. DS-GNN methods all achieve improvements over base models (indicated by the bold numbers), e.g., in CiteSeer, DS-GCN (79.9%) achieves large improvement over GCN (75.9%). Furthermore, applying our proposed method to the base GNN models can achieve the better performance, outperforming the state-of-the-art models. Compared with DropEdge which takes GCN as base model, our DiverseSample (DS) mechanism achieves better or comparable results.

**Analysis of Sampling Numbers $K$**

To analyze the impact of sampling times $K$, we show the accuracy of Cora with different $K$ in Fig. 3. It is observed that the accuracy increases as $K$ becomes larger at the beginning. This is owing to the increasing probability of achieving injectivity. When $K$ continues to increase, the injective probability becomes stable and close to 1, resulting in the accuracy also showing a stability. The results show that we can achieve better experimental results with a small $K$, e.g., $K = 5$ in Fig. 3), which proves the practicability of our method.

**Analysis of Sampling Strategy**

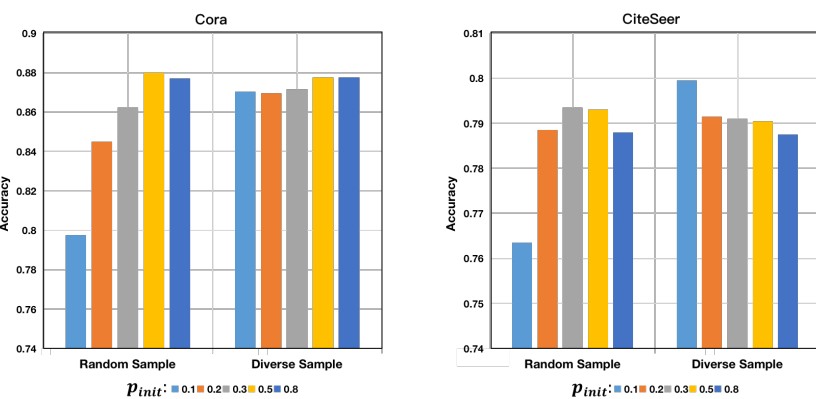

Figure 4: Diverse sampling VS Random sampling

To evaluate the effectiveness of the proposed diverse sampling strategy, we compare it with random sampling strategy. Fig. 4 show the accuracy of Cora and CiteSeer with different initial node sampling probability $p_{init}$ respectively when sampling times $K = 5$. The results demonstrate that: 1) the performance of random sampling will be significantly affected by $p_{init}$. The model performs poor with a relatively small $p_{init}$, while the model effect is significantly improved when $p_{init}$ increases. Thus, random sampling requires a relatively large $p_{init}$ to maintain the sampling effect and model

performance. 2) The performance of Diverse sampling is relatively stable, obtaining comparable performance even with a small $p_{init}$. This is owing to diverse sampling considers the three guidelines mentioned earlier, which proves the effectiveness of the diverse sampling strategy we proposed.

Furthermore, under diverse sampling, we observe that the best $p_{init}$ are different among datasets. If better results achieved under a smaller $p_{init}$, our diverse sampling can achieve improvements over random sampling. Also, it is observed that Cora performs well with a larger $p_{init}$, while CiteSeer achieves better performance when $p_{init}$ is smaller, i.e., Cora tends to keep more edges in sampled graphs. Similarly, PubMed also performs better when $p_{init}$ is smaller. The reason may be related to the property of label smoothness "$\lambda$" proposed in CS-GNN (Hou et al., 2019), which measures the quality of surrounding informations. A larger $\lambda$ implies that nodes with different labels are connected in the graph. In CS-GNN, Cora has a smaller $\lambda = 0.19$, while Citeseer with a larger $\lambda = 0.26$ as well as PubMeb with $\lambda = 0.5$. These values indicate that the surrounding informations are of higher quality in Cora, which explain why Cora prefers to keep more edges.

### 4.4 NODE-BASED MULTI-LABEL CLASSIFICATION

To enrich the type of node-level tasks, we collect a new dataset from the DouBan website and offer an multi-label classification task. Each sample in this dataset has its descriptions (features), genres (labels), and similar movies (edges), as illustrated in Fig. 5. These similar movies (edges) are directly provided by DouBan website based on user co-preference. This dataset has 28 labels and each movie may have more than one label. We define the task as tagging the movie with its own labels. The last row in Table 1 shows an overview of the newly collected DBMovie dataset.

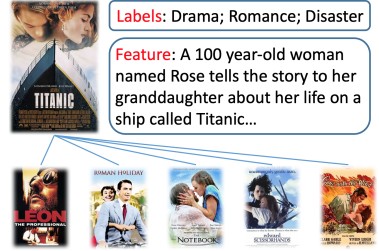

Figure 5: One Sample of DBMovie

| Method | MAP | F1@3 | NDCG@3 |
|--------|-----|------|--------|
| MLP | 54.9% | 39.4% | 50.6% |
| DeepWalk | 61.6% | 44.7% | 59.1% |
| GCN | 83.2% | 60.2% | 82.1% |
| **DS-GCN** | **83.6%** | **60.5%** | **82.6%** |
| GAT | 83.3% | 60.5% | 82.6% |
| **DS-GAT** | **84.0%** | **61.0%** | **83.0%** |

Table 3: Results of Node-based Multi-label Classification

**Performance on Node-based Multi-label Classification**

We now validate the effectiveness of DS-GCN and DS-GAT. For node-based multi-label classification, we leverage the widely used ranking metrics to evaluate our method, including Mean Average Precision (MAP), F1, and Normalized Discounted Cumulative Gain (NDCG). These metrics encourage the correct label to be placed ahead of irrelevant labels, and a larger value indicates better performance. Experimental results on DBMovie are reported in Table 3. In the multi-label classification task, GNN models, i.e., both GCN and GAT, also perform much better than traditional methods, including MLP (only using attributes) and DeepWalk (only using only). GAT performs better than GCN, owing to its attention mechanism to learn edge weight. Under all evaluation metrics, our proposed method achieves consistently an improvement over GCN and GAT, showing an increased expressive power of model.

## 5 CONCLUSION

We proposed DS-GNN to improve the expressive power of graph neural networks. Different from previous methods that aim to implement injectivity via designing complicated layer-wise aggregation functions, we focus on designing the diverse rooted sub-graphs for each node. To enhance the diversity of rooted sub-graphs, we design the diverse sampling strategy. As the number of samples increases, we can achieve injectivity with higher probability. Extensive experiments on node-based multi-class classification and node-based multi-label classification prove that our method can achieve improvements over the baselines. In the future, we consider to extend our method to graph classification scenarios.

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

## A    DESCRIPTION OF BASE MODELS

The two basic models Graph Convolutional Networks (GCN) and Graph Attention Networks (GAT) output the probability distributions $Z$ over labels for all the nodes. Formally, the probability distributions are calculated as follows

$$Z = f(X, A) = \text{softmax}(\hat{A}\text{ReLU}(\hat{A}XW^{(0)})W^{(1)}), \tag{11}$$

where $W^{(0)} \in \mathbb{R}^{p \times d}$ and $W^{(1)} \in \mathbb{R}^{d \times m}$ are weight matrices with $d$ as the dimension of the hidden layer. The first layer leverages rectified linear unit (ReLU), and the second layer leverages softmax to obtain a probability distribution $Z \in \mathbb{R}^{n \times m}$. $Z$ is a row-normalized matrix, where each row represents the probability that the node belongs to corresponding labels.

In GCN, $\hat{A} = D^{-\frac{1}{2}}\tilde{A}D^{-\frac{1}{2}}$ is the symmetric normalized adjacency matrix with $\tilde{A} = A + I$, and $I$ is the identity matrix. $D \in \mathbb{Z}^{n \times n}$ is the diagonal degree matrix with $D_{ii} = \sum_j \tilde{A}_{ij}$.

While, GAT modifies $\hat{A}$ by introducing the attention mechanism:

$$\hat{A}_{ij} = \frac{\exp(\text{LeakyReLU}(a^T[h_i||h_j]))}{\sum_{k \in N(v_i)} \exp(\text{LeakyReLU}(a^T[h_i||h_k]))}, \tag{12}$$

where $N(v_i)$ is the set of neighboring nodes of node $v_i$, $a \in \mathbb{R}^{2d}$ is the attention vector, $h_i \in \mathbb{R}^d$ is the representation of the $i$-th node, and $||$ is the concatenation operator.

## B    THE DIFFERENCES BETWEEN NODE SAMPLING AND OTHER RELATED CONCEPTS

We contrast the differences between node sampling and other related concepts including dropping features and dropping edges. The Dropout (Srivastava et al., 2014) is proposed to prevent over-fitting by randomly dropping features. Dropping edges can be regarded as a generation of dropout from dropping features to dropping edges. DropEdge (Rong et al., 2019) proposed randomly setting the elements in the adjacency matrix to be zeros. Also, some recent methods, including Graph-SAGE (Hamilton et al., 2017), ASGCN (Huang et al., 2018), can be seen as dropping edges. These methods extend GNN to large graphs by sampling neighbors for each node, i.e., sampling a neighbor is equivalent to sampling an edge. Hasanzadeh (Hasanzadeh et al., 2020) propose a general framework for these methods. However, for our method, dropping edges may cause that two nodes who are the same with identical neighbors obtain different representations, since dropping edge cannot guarantee that these two nodes still have identical neighbors after edge sampling. Thus we leverage node sampling, ensuring that the nodes with the same input features and identical neighbors will still have the same representation.

## C    PROOF OF THEOREM

**Theorem** As the number of samples increases, the probability that the multi-sets for two nodes with different attributes or different neighbors are different can be close to 1.

**Proof**: Let $p$ denote the probability that representations of node $u$ and node $v$ are same on the $i$-th sampled graph, i.e.,

$$p = \text{Probability}(u_i = v_i). \tag{13}$$

Let $q$ denote the the probability that representations of node $u$ and node $v$ are same on two different sampled graphs, i.e.,

$$q = \text{Probability}(u_i = v_j) \quad for \quad i \neq j. \tag{14}$$

It's reasonable to assume that such a probability based on different sampled graphs is smaller than base on the same sampled graph, i.e., $q <= p$.

For the two multi-sets, without losing generality, we can sort the element in each multi-set by descending order based on a specified uniform ordering rules. After sorting, the two multi-sets can be written as

$$\text{Multi} - \text{set}(\hat{u}) = \{u_{i_1}, u_{i_2}, \cdots, u_{i_K}\}, \qquad \text{Multi} - \text{set}(\hat{v}) = \{v_{j_1}, v_{j_2}, \cdots, v_{j_K}\}, \tag{15}$$

where $i_k$ represent the original index of $u_{i_k}$ in $\mathrm{Multi-set}(u)$, so as $j_k$. For these two sorted multi-sets, if and only if their elements be equal at each position, these two sorted multi-sets can be the same, otherwise they are different.

As we defined, the probability of "$u_{i_k} = v_{j_k}$" is smaller than $p$, thus the probability that two multi-sets are the same can be defined as following based on Bayesian Equation:

$$\mathrm{probability}(\mathrm{Multi-set}(\hat{u}) = \mathrm{Multi-set}(\hat{v})) \leq$$
$$p \times p(u_{i_2} = v_{j_2} | u_{i_1} = v_{j_1}) \times \cdots \times p(u_{i_K} = v_{j_K} | u_{i_1} = v_{j_1}, \cdots, u_{i_{K-1}} = v_{j_{K-1}}). \tag{16}$$

Therefore, the probability that the injectivity can be achieved between the two multi-sets is larger than $1 - \mathrm{probability}(\mathrm{Multi-set}(\hat{u}) = \mathrm{Multi-set}(\hat{v}))$. Since $u$ and $v$ have different attributes or neighbors, each item in Eq. 16 can be smaller than 1. Thus, with large number of samplings, we can achieve injectivity with a high probability.

## D STANDARD DEVIATION OF NODE-BASED MULTI-CLASS CLASSIFICATION

Table 4: Results with Standard Deviation of Node-based Multi-class Classification

| Method | Cora | CiteSeer | PubMed |
|---|---|---|---|
| DS-GCN | **88.0**%±0.36% | **79.9**%±0.23% | **90.5**%±0.19% |
| DS-GAT | **88.2**%±0.31% | **80.0**%±0.26% | **91.0**%±0.29% |

