# OpenReview forum: "Towards Powerful Graph Neural Networks: Diversity Matters"
_ICLR.cc/2021/Conference — Reject_

### Official Review · AnonReviewer2 · 2020-10-28
**This paper introduces a graph sampling method to increase the chance that GNN can differentiate different node representations. The idea makes sense but is straightforward. The novelty is limited.**

**Rating:** 4
**Confidence:** 5

**Review:**

In this paper, the authors propose to sample nodes of a given graph multiple times to form a set of K sub-graphs. GNNs are then applied on each sampled graph for learning node representations. For each node, all representations are combined for the downstream tasks. The idea of doing multiple sampling is to increase the chance that nodes with different neighborhoods can be more different in the set of sampled graphs, than only considering the original single graph. In contrast, nodes with same neighborhood will remain the same over all sampled graphs. This mechanism helps discriminate node representations better.

The key technical contribution of the paper is the proposed sampling strategy, which, with several proposed guidelines, is designed to reduce the possible number of samples K. The strategy is mostly an empirical design. It is good to see some analysis on how do the sample times affect the difference of node representations, but it seems to be very intuitive.

As such, the technical contribution of this paper is limited. The idea to use multiple samples to improve performance seems also a straightforward way to improve the coverage of nodes' local structures.

In the experiments, it is interesting to compare with GraphSage by sampling nodes' neighborhoods multiple times. Perhaps with a small number of sampling times, GraphSage's performance can be further improved. Especially, its performance in Table 2 is close to the proposed method.

In Fig. 3, the decrease of accuracy with increased K remains to be analyzed. Intuitively, the performance should be stable as K increase.

From Fig. 4, the improvements brought by the proposed sampling strategy is small compared with the random sampling, which questions the importance of the proposed strategy. In addition, the results cannot show how does the proposed strategy maintain superior performance with a smaller K compared with the random sampling.

In Table 3, it is not clear on why do the three GNN methods in Table 2, GIN, GraphSage and DropEdge-GCN are excluded. It is better to make all comparison consistent.

---

### Official Review · AnonReviewer4 · 2020-10-28
**The paper presents a method for subgraph sampling which is interesting, but the theoretical and empirical aspects should be improved**

**Rating:** 4
**Confidence:** 4

**Review:**

The paper presents a method that should increase the expressive power of GNN. This method includes sampling subgraphs out of the input graph using a novel diverse sampling method and calculating the output of each node by using a shared GNN on each sampled graph and summing over the outputs. The empirical evidence presented shows that this method outperforms other GNN architectures such as GCN and GAT on several node classification tasks.

I think the diverse sampling (DS) method is interesting and well-motivated so that the sampled subgraphs are diverse, each node appears at least once and high degree nodes have a higher probability to appear in each sample.  The DS method is also applicable to any GNN architecture which is an advantage. The experiments show that a 2-layer DS combined with GCN or GAT improves on the base models with the DS. Another contribution of this paper is presenting a new dataset, namely DBMovie, and showing that DS slightly improves performance on this dataset compared to the base models.

I have a couple of concerns regarding this paper, especially the theoretical grounding, whether the model is invariant, and the experimental setup. In more details:
1)	 The theorem (page 6) is written in a very informal manner, and I am concerned it doesn’t show that using DS actually increase the expressive power of GNN: stating that a probability “can be close to 1” is very unclear, it can also be *not* close to 1, so in this sense, the theorem doesn’t state anything.
2)	I have some concerns about the correctness of the proof of this theorem, as there are a couple of handwaving arguments there (appendix C). Some examples: “It’s reasonable to assume that such a probability based on different sampled graphs is smaller than the base on the same sampled graph, i.e., q <= p.”, this statement is unclear. In order to claim this, p and q should be exactly calculated, they might depend on the structure of the graph, on the nodes that are chosen, etc. In equation (16), it is not clear why each item is smaller than 1, and even if each item is smaller than 1, still each item can be arbitrarily close to 1 so that this probability might be very large.
3)	The authors claim that the DS method improves the expressive power of GNNs, while the theorem only states (informally) that w.h.p two nodes with different attributes can be distinguished. On the other hand, it is not clear why two nodes with the exact same neighborhood will have the same output. I am concerned that using this sampling method might make GNNs not invariant – so that two nodes with the exact local structure will have different outputs. This might be OK in general if the probability for this to happen is small (which is believe is the case here) but should be shown formally, or at least discussed.
4)	My main concern about the experiment is that they were done with a very shallow GNN (2-layer), which represents a very small neighborhood of each node. I think it is important to test deeper GNN in order to show the robustness of the DS method.
5)	The paper is missing a related work section, so it is hard to situate the novelty of the sampling method compared to other graph sampling methods that are also used in modern GNNs. Some examples:
https://arxiv.org/abs/1907.04931
https://arxiv.org/abs/1806.01973
https://arxiv.org/abs/1801.10247
https://arxiv.org/abs/1905.07953
Also, since it is claimed that DS-GNN increases expressive power, there should be a clearer discussion about works related to the expressive power of GNN (there are many such recent works), the only relevant paper mentioned is Xu et al 2018.
6)	The new DBMovie dataset that is presented is not clear enough. What are the features of each node? According to Figure 5, this is a sentence representing each movie, so how each sentence is represented as a node feature? It would also be beneficial to state how many movies (nodes) are there of each genre (label). Lastly, I highly advise the authors to upload this dataset as supplementary material for the paper, so it would be possible to verify the results.

Some minor typos:
-	Page 4, second paragraph, 5th sentence from the end “when its degree”
-	Section 4.4 “an multi-label” -> “a multi-label”
-	Section 4.2 second sentence “we choose the the”
-	Section 4.4 “DeepWalk (only using only)”

To conclude, although the DS method seems interesting, the theoretical aspect of the paper is not clear enough, more experiments should be done with deeper networks (more than 2 layers), and a clear related work should be added in order to situate more clearly this method with respect to other known methods.

---

### Official Review · AnonReviewer1 · 2020-10-29
**A good but not self-justified attempt**

**Rating:** 4
**Confidence:** 4

**Review:**

This paper claims that existing GNNs often suffers from the limited capability of the aggregation function. This paper proposes a new framework of a diverse sampling of the graph to solve this problem. Specifically, this paper first samples several different graphs and use GNN on each graph to generate features, and finally use a type of injective multi-set aggregation function to obtain the final representation. The experimental results show that adding this module to GCN and GAT can further boost node-based multi-class classification performance.

Overall, I think it is crucial to design how to choose neighbors and aggregate the neighbors' information. However, I did not find theoretical support or scenario support for the proposed diverse sampling method.

Concerns:
1. The problem that the authors focus on is interesting. Different graphs can result in different features of one node. However, I did not find theoretical support of the method design (like [1]) or scenarios to illustrate why we need to subsample the graphs.
2. The sampling methods are also designed based on intuitions without further theoretical analysis of the sampling method's design.
3. GAT can also learn different weights on different neighbors, which can be regarded as adaptively and softly select the neighbors. Could the author further illustrate why GAT cannot capture the features that the proposed method can? In other words, what is the fundamental difference?
4. The Theorem in this paper show that the probability that the multi-sets for two nodes with different attributes or different neighbors are different can be close to 1. As I understand it, [2] can also achieve this goal. Can the authors provide theoretical and experimental comparison?
5. Diversity may also lead to the weakening of generalization, could the author further claim on that point?
6. Important baselines are missing. How to sample informative neighbors to improve GNN has been widely studied. For example, Chen et al. proposed a node importance sampling algorithm [3], which has been proven effective in accelerating model convergence. These closed related baselines should be carefully discussed and evaluated in experiments.  Besides, the performance gain over vanilla GNN is tiny in many cases. More importantly, Figure 4 shows that diverse sampling does not outperform random sampling when the initial sampling probability is reasonably large.

[1] Xu, Keyulu, et al. "How powerful are graph neural networks?." arXiv preprint arXiv:1810.00826 (2018).

[2] You, Jiaxuan, Rex Ying, and Jure Leskovec. "Position-aware graph neural networks." arXiv preprint arXiv:1906.04817 (2019).

[3]. Chen J, Ma T, Xiao C. Fastgcn: fast learning with graph convolutional networks via importance sampling[J]. arXiv preprint arXiv:1801.10247, 2018.

---

### Official Review · AnonReviewer3 · 2020-11-02
**Official Blind Review #3**

**Rating:** 4
**Confidence:** 5

**Review:**

This paper presents diverse sampling to improve the expressive power of GNNs.The key idea is to construct diverse rooted sub-graphs for each node and obtain the target node representation by aggregating the representation of diverse neighborhoods. The experiment results on 4 different datasets justified the effectiveness of the proposed method.

Strengths:
+ This paper is well written and organized.
+ Enhancing the diversity of neighborhood sampling is an interesting idea to investigate in GNNs

Weaknesses:
- It is not clear whether this idea can generalize to other GNNs, e.g., ResGCN, IncepGCN, etc.
- No theoretical justification is provided for the proposed diverse sampling.
- Some details of the proposed technique are missing.

Questions:

1. What’s the relationship between the proposed technique and graph sparsification or DropNode?

2. Why are GraphSAGE and DropEdge GCN not compared for multi-label classification?

3. In Table 2, the performance improvement over baseline is limited. It is difficult to determine whether the performance
improvement really comes from the diverse sampling strategy.

4. What’s the computational complexity of GCN based on a diverse sampling strategy?

---

### Official Review · AnonReviewer5 · 2020-11-06
**Review Comment #5**

**Rating:** 3
**Confidence:** 3

**Review:**

**Review Summary**

I have a question about the correctness of this paper because I do not think the statement and proof of the main theorem are mathematically rigorous nor correct. Regarding empirical evaluation, I think this paper correctly evaluates the proposed diverse sampling enhances the performance of base GNNs in the multi-class and multi-label node classification problems. However, I have several questions about the discussion about the ablation studies.

**Summary**

This paper proposed DS-GNN, a GNN augmented with plug-in diverse sampling methods. It conducts node sampling to construct multiple subgraphs and apply base GNNs to them. Theoretically, they claimed to show that DS-GNN has high expressive power when the number of sampled subgraphs are sufficiently large. Empirically, they applied DS-GNN to multi-class and multi-label node classification problems and claimed to demonstrate that the proposed sampling method enhances the performance of base GNNs and that DS-GNN is comparable to the state-of-the-art GNN models. Finally, this paper constructs a new dataset, DBMovie, which can be used for multi-label node classification problems.

**Claim**

If I understand correctly, this paper's claims are as follows:

- [1-1] Claim 1: The proposed DS-GNN has theoretically powerful expressive power.
- [1-2] Claim 2: The proposed DS-GNN empirically performs well.
- [1-3] Claim 3: Practically, DS-GNN applies to any GNNs.

**Soundness of the claims**

Can theory support the claim?

- [2-1] Claim 1: This paper justified this claim by proving that the proposed method is injective in the sense that "any two nodes that have either different features or different neighborhood structures are mapped to different values" (I call this property as Property A in this review). I agree with this paper in that it is one way to prove the expressive power of GNNs. However, I question the correctness of the main theorem for establishing this result (see Correctness section).
- [2-2] Claim 2 is not a theoretical one.
- [2-3] The model described in Section 3 supports Claim 3.

Can empirical evaluation support the claim?

- [3-1] Claim 1 is not an empirical one.
- [3-2] Claim 2: Experiments in Tables 2 and 3 support this claim for node classification tasks. The ablation studies in Figures 3 and 4 are appropriate to confirm if the proposed sampling strategy works as expected.
- [3-3] (This is not a requirement but a suggestion) I think it would be beneficial to claim the proposed method's strength if this paper compares it experimentally with CPNGNN and PNA, which the authors mentioned in the introduction. By doing so, I think this paper could claim that the tricky design (according to this paper) of CPNGNN and PNA is harmful to empirical performance.
- [3-4] Claim 3: They applied the proposed diverse sampling to two types of GNNs: GCN and GCT, which empirically support Claim 3.

**Significance and novelty**

Novelty

- [4-1] This paper has constructed a new dataset, DB-Movie, for multi-label node classification problems.

Relation to previous work

- [5-1] I think this paper discuss the relation with previous work well.
  - [5-1-1] The idea of sampling the input graph itself has been employed in existing studies, such as DeepWalk, DeepEdge, and GraphSage. The proposed method differs from them because they are random-walk-based (DeepWalk) or edge-sampling-based (DeepEdge and GraphSage), while DS-GNN is node-sampling-based. I think it is better to mention FastGCN [Chen et al. 2018], another edge-sampling-based GNN.
  - [5-1-2] Designing GNNs that have the maximal expressive power has been investigated by previous studies, such as CPNGNN and PNA. The proposed method took a strategy different from these methods.

[Chen et al. 2018] Chen, J., Ma, T., & Xiao, C. FastGCN: Fast Learning with Graph Convolutional Networks via Importance Sampling. ICLR2018.

**Correctness**

Is the theorem correct?

- [6-1] If I understand correctly, the statement nor proof of the main Theorem is not appropriate mathematically.

  - [6-1-1] I am not confident I understand what "can be close to 1" correctly. Does it mean that the probability tends to 1 as some parameters (such as K) goes to infinity? I think "can be close to 1" in the statement is not a popular mathematical term, so I would recommend writing what it means mathematically. Also, the term "with high probability" in the proof has several meanings depending on the context. So, it is better to clarify what it means mathematically.
  - [6-1-2] Assuming that my understanding of "can be close to 1" is correct (the limit is in terms of $K\to \infty$), I think the proof does not prove it. Specifically, the proof claims that from the assumption that "$u$ and $v$ have different attributes or neighbors, each item in Eq. 16 can be smaller than 1.", it is true that the injectivity holds with high probability. However, it is not true in general that $0\leq a_i < 1$ for all $i$ implies $\prod_{i=0}^\infty a_i = 0$. ~~For example we have $(1-1/n)^n→e^{-1} > 0$ as $n\to \infty$.~~  [Nov. 11, 2020 edit]: Since I found this example is not appropriate, I gave another example. See the reply to this comment.
  - [6-1-3] $p$ in (13) implicitly depends on the index $i$. Does (13) mean that the probability is the same for all $i$ ? Same is true of (14), in which $q$ implicitly depends on $i$ and $j$.
  - [6-1-4] The proof says that "it is reasonable to assume that [...], $q\leq p$". I recommend explictly writing the assumption not in the proof but in the statement.
- [6-2] Even if the main Theorem is true, I think we cannot show that DS-GNNs has the Property A. If I understand correctly, the proof assumes that the probability that $h_u^{i, L} = h_v^{i, L}$ is not 1 for some $i$, where $h_\ast^{i, L}$'s are the outputs of the base GNN. In order to prove that DS-GNN has Property A, this paper seems to assume implicitly the following assumption (Assumption B) and apply the main Theorem. However, I think Assumption B is almost equivalent to assuming that the base GNN is sufficiently strong, which is what we want the desired GNN to have.
    - [6-2-1] Assumption B: The base GNN can output different representations with non-zero probability if $u$ and $v$ has either different features or different neighborhoods.

Is the experimental evaluation correct?

- [7-1] Multi-class node classification problems in Section 4.3
  - [7-1-1] Table 2: Basically, OK. However, I wonder why only DS-GCN and DS-GAT have standard deviations (Table 4), while others not.
  - [7-1-2] Figure 3: This paper discussed that accuracy is stabilized when the injectivity probability is close to 1. However, I have several questions about this statement. First, the accuracy in Figure 3 is not stabilized: the performance peaks at K=5 and drops at K=6. Second, since this experiment does not measure the injectivity probability, we cannot say the injectivity probability is truly close to 1. Finally, we do not know that the injectivity probability is truly correlated with performance accuracy.
  - [7-1-3] Figure 4: This paper discussed that "[Furthermore], under diverse sampling, we observe that the best $p_{init}$ are different among datasets." However, we cannot check whether it is correct because the concrete values of $p_{init}$ are not shown. I would recommend this paper to show the values of $p_{init}$ if this paper wants to claim this sentence.

- [7-2] Multi-label node classification problem in Section 4.4

  - [7-2-1] I did not find any critical problems. I think it was good that this paper compared the proposed method with not only GNNs but also MLP and DeepWalk to show that the task needs both node features and graph topologies.

Is the experiment reproducible

  - [8-1] Experiment code is not provided. However, hyperparameters and train/validation/test splits are written.
  - [8-2] Since the DBMovie is not available, we cannot reproduce the results in Section 4.4.

**Clarity**

Can I understand the main point of the paper easily?
- [9-1] Yes. However, the main theorem and its proof is not clear.

Is the organization of paper well?
- [10-1] Yes

Are figures and tables appropriately made?
  - [11-1] Yes

**Additional feedback**

- [12-1] It is better to discuss the computational complexity of the proposed methods. If I understand correctly, the time and space complexity is multiplied by K. Is my understand correct?
- [12-2] The new dataset, DBMovie, is interesting. Do you have a plan to publish the dataset?

Minor comments

- [13-1] Page 1, Abstract: L.10: For a target node, diverse sampling offers it diverse neighborhoods, i.e., [...] → For a target node, diverse samling offers diverse neightborhoods to it., i.e., [...]
- [13-2] Page 2, Paragraph 3: For convenience, we denote with DS-GNN ... → we denote by DS-GNN
- [13-3] Page 3, L.3: Remove DS-GNN
- [13-4] Page 4, paragraph after (4): I could not understand what this paper intended to mean by "a generation of Dropout from dropping features to dropping edges"
- [13-5] Page 6, Section 4.1, Note that [...] 10 sampled garphs [...] → graph
- [13-6] Page 6, Section 4.1, All GNN models [...] leverage Relu as [...] → ReLU
- [13-7] Page 8 Section 4, Last Paragraph: [...] Deep Walk (only using only) [...]: → only using graph structures
- [13-8] K is not writen in italic in some places (e.g., Page4 Sampling Strategy, last sentence)
- [13-9] Appendix, after (14): $q <= p$→$q\leq p$

---

> ### Comment · AnonReviewer5 · 2020-11-11
> **Change of counterexample**
>
> First, I have added indices [X-Y] and [X-Y-Z] to my review comments to refer my comments easily.
>
> I found that the counterexample I had given was not appropriate. So, I change the counter example as shown below. I am sorry for my mistake. Even this change, my opinion nor evaluation score do not change.
>
> Specifically, the example in [6-1-2] was not appropriate as a counterexample of the claim that $0\leq a_i < 1$ does not imply $\prod_{i=1}^\infty a_i = 0$. So, I want to give another example: $\prod_{i=1}^\infty  (1-1/2^n) = (1/2; 1/2)_\infty \approx 0.289$, where $(a; q)_k$ is the q-Pochhammer symbol (https://mathworld.wolfram.com/q-PochhammerSymbol.html) .

---

### Decision · Program_Chairs · 2021-01-07
**Final Decision**

**Decision:**

Reject

**Comment:**

The reviewers, including me, agreed that considering sampling diversity is interesting and reasonable when designing GNNs. However, the proposed method is too heuristic and empirical. Without the authors' feedback, I tend to reject this work.